# Magnetoelectric Magnetic Field Sensors: A Review

**DOI:** 10.3390/s21186232

**Published:** 2021-09-17

**Authors:** Mirza Bichurin, Roman Petrov, Oleg Sokolov, Viktor Leontiev, Viktor Kuts, Dmitry Kiselev, Yaojin Wang

**Affiliations:** 1Institute of Electronic and Information Systems,Yaroslav-the-Wise Novgorod State University, ul. B. St. Petersburgskaya, 41, 173003 Veliky Novgorod, Russia; roman.petrov@novsu.ru (R.P.); oleg.sokolov@novsu.ru (O.S.); viktorsergeevich.novsu@gmail.com (V.L.); 2Department of Materials Science of Semiconductors and Dielectrics, National University of Science and Technology MISiS, Leninskiy Prospekt 4, 119049 Moscow, Russia; viktor.kuts.3228@yandex.ru (V.K.); dm.kiselev@gmail.com (D.K.); 3School of Materials Science and Engineering, Nanjing University of Science and Technology, Nanjing 210094, China; yjwang@njust.edu.cn

**Keywords:** magnetoelectric sensors, magnetoelectric effect, magnetostrictive component, piezoelectric component, composites, magnetostriction, piezomagnetic coefficient, magnetoelectric voltage coefficient, magnetic sensitivity, noise level

## Abstract

One of the new materials that have recently attracted wide attention of researchers are magnetoelectric (ME) composites. Great interest in these materials is due to their properties associated with the transformation of electric polarization/magnetization under the influence of external magnetic/electric fields and the possibility of their use to create new devices. In the proposed review, ME magnetic field sensors based on the widely used structures Terfenol—PZT/PMN-PT, Metglas—PZT/PMN-PT, and Metglas—Lithium niobate, among others, are considered as the first applications of the ME effect in technology. Estimates of the parameters of ME sensors are given, and comparative characteristics of magnetic field sensors are presented. Taking into account the high sensitivity of ME magnetic field sensors, comparable to superconducting quantum interference devices (SQUIDs), we discuss the areas of their application.

## 1. Introduction

ME composites are functional materials that are created on the basis of various physical effects and, along with semiconductor materials, determine the progress of electronic materials science today. The ME effect in composites, as is known, consists in the induction of electric polarization in an applied external magnetic field or, conversely, in magnetization by an external electric field, and is the result of the interaction of the electric, magnetic, and elastic subsystems of the composite. Compared to single-phase materials, composites show a giant ME response at room temperature and are ready for practical applications. The required high values of the ME coefficients can be obtained by choosing components with high piezoelectric and piezomagnetic modules. At present, a large number of works are known that are devoted to the wide study of ME composites with the aim of their application in magnetic field sensors [1,2,3,4,5,6,7,8,9]. The purpose of this review is to compile a collection of the best works on the theory and application of the ME effect for creating magnetic field sensors.

The first report on the ME magnetic field sensor was made at the International Conference, MEIPIC-4 in 2001 [10]. The sensor was a disk or plate based on a bulk composite of the composition of iron-yttrium garnet–zirconate–lead titanate (PZT) and a layered structure based on nickel-zinc ferrite—PZT. The maximum sensor sensitivity was of 0.06 mV/Oe.

Further, of note are the results of the group of Prof. D. Viehland, who significantly advanced our knowledge of ME magnetic field sensors and their capabilities [11,12,13,14,15,16,17,18,19]. In [11], Dong et al. showed that L-T mode in the structure Terfenol-PMN-PT has extremely high magnetic field sensitivity: at room temperature, an output ME voltage with a good linear response to magnetic field H_ac_ was found over the range of 10^−11^ < H_ac_ < 10^−3^ T. Then, a number of articles was published in which the authors of this group applied new ideas: the use of push–pull [12] and bimorph structures [13] to increase a ME effect and sensitivity by reducing the vibrational and thermal noises [14], and the transition to higher signal processing frequencies [15,16], which made it possible to significantly increase the sensitivity of sensors and achieve a pico-Tesla sensitivity of a sensor at room temperature [17,18,19]. These innovations made it possible to improve the parameters and obtain high values of the sensitivity and noise characteristics of sensors both in the low-frequency region and in the field of electromechanical resonance (EMR). The sensitivity limit was about 20 pT/Hz^1/2^ or 2 × 10^−16^ C/Hz^1/2^ at 1 Hz, and about 2 fT/Hz^1/2^ at 78 kHz in the EMR range [17]. Extremely low values of the equivalent magnetic noise of 6.2 pT/Hz^1/2^ at a frequency of 1 Hz were obtained on a heterostructure consisting of a PMN-PT bimorph doped with 1% Mn and longitudinally magnetized Metglas layers. As the frequency increased to 10 Hz, the equivalent magnetic noise significantly decreased and amounted to <1 nT/Hz^1/2^ [3,19]. A sharp jump in the number of articles on ME magnetic field sensors was noted after the publication of a number of works in which a sensitivity of 1–10 pT was achieved at room temperature [4,5,6,7,9].

As known, most of the studies of ME magnetic field sensors were carried out on the basis of such components as piezoelectric PZT, PMN-PT, lithium niobate and magnetic Terfenol, and Metglas in the linear ME effect mode. At the same time, it should be noted that in recent years, original works have appeared in which new approaches were used and high characteristics of sensors were obtained. In particular, the magnetostrictive component was replaced by a NdFeB magnet [20], then a miniature and highly sensitive MEMS structure [21,22], and NEMS and nanoresonator [23,24,25] for biomedicine were proposed, and a nonlinear regime and new components such as piezoelectric AlN and soft magnetic Ni [26,27,28] were used. In addition, new possibilities for developing sensors on the basis of the ΔE—effect have been discussed [29,30]. In this article, we look at these options in more detail.

Analysis of the characteristics of ME magnetic field sensors shows that in a number of parameters they are superior to the commonly used Hall and magnetoresistive sensors and have great prospects for application in technics and biomedicine. With this in mind, this article aims to review the current state of the art of ME magnetic field sensors. The article is organized as follows. The second section provides information about the principle of operation and topology of ME sensors developed on the basis of the structures Terfenol-PZT/PMN-PT, Metglas-PZT/PMN-PT, Metglas-Lithium Niobate, and other original structures with the best characteristics. Special attention is paid to promising ME sensors presented in recent years. The third section compares the features and discusses how they can be improved. Finally, some concluding remarks close the article.

## 2. Main Section

As already noted, this part of the article will provide information on the ME magnetic field sensors based on Terfenol –PZT/PMN-PT, Metglas PZT/PMN –PT, Metglas –Lithium Niobate, and other original structures, on which high characteristics were obtained.

### 2.1. Terfenol—PZT/PMN-PT

Modern magnetic field sensors today need to measure magnetic fields of 10^−12^ T and below at a low frequency of 10^−3^ to 10^2^ Hz. In addition, new magnetic field sensors must operate at room temperature, as well as be relatively small and passive. The most sensitive magnetic field sensor today is a SQUID, the sensitivity of which reaches 5 × 10^−18^ T/Hz^1/2^. However, in order to achieve such a gigantic sensitivity, a number of conditions must be met: (i) a shielded room, and (ii) cryogenic cooling. As can be seen, the SQUID magnetometer has a number of significant drawbacks that complicate its widespread use.

As a result of recent research on sensors based on ME composites, it has become clear that they have enormous potential for detecting weak magnetic fields. Applying the lock-in amplifier method, Dong et al. showed the possibility of detecting a magnetic field of the order of 10^−12^ T at a frequency of *f* > 1 Hz. The magnetic field sensor based on ME composites is a small passive device that operates at room temperature and has a sensitivity at low frequencies in the picotesla range. The main task of the studies carried out by various scientific groups is to obtain the maximum ME voltage coefficient, increase the sensitivity, and obtain the minimum noise of the ME magnetic field sensor.

As noted in the introduction, the Virginia Tech group made the main contribution to the study of ME sensors based on PZT/PMN-PT-Terfenol-D structures. In [31], Dong et al. used a layered ME composite in the form of a ring, consisting of a Pb(Zr,Ti)O_3_(PZT) piezoelectric layer and a magnetostrictive Terfenol-D (Tb_1−x_Dy_x_Fe_2−y_) layer, as shown in Figure 1. This ME structure consists of circularly magnetized Terfenol-D layers and a circularly polarized piezoelectric PZT layer, C-C mode layered structure.

This mode allows for the increase in the sensitivity of the sensor, due to the amplification of deformations/vibrations around the circumference in the composite.

As a result, a sensitivity of the order of 10^−9^ T was achieved, and the maximum amplitude response was 260 mV at an alternating field H_ac_ = 1 Oe in the frequency range from sub-Hz to kHz. However, the main disadvantage of such a sensor is a large displacement field, of the order of H_dc_ = 500 Oe.

In 2006, Dong et al. [32] found that small rectangular ME composites consisting of the magnetostrictive material Terfenol-D and the piezoelectric PZT are very sensitive to small changes in the constant magnetic field. The sensitivity limit is H_dc_ < 10^−3^ Oe (10^−7^ T) using a constant amplitude low frequencies drive, and H_dc_ < 10^−4^ Oe (10^−8^ T) under resonant drive.

The proposed sensor of a constant magnetic field consists of an L-T mode ME composite and a coil wound on it for passing an alternating current I_ac_, as shown in Figure 2.

The authors showed that as a result of the detection of small constant magnetic fields by the structure Terfenol-D–PZT-Terfenol-D, the alternating voltage changed across the PZT layer. In this case, the dV_ME_/dH_dc_ ratio increased by more than five times at the electromechanical resonance frequency, which made it possible to achieve a sensitivity of up to 10^−8^ T. They indicated the feasibility to detect DC magnetic field variations in the 10^−9^ T, and also suggested that it is possible to obtain a higher sensitivity by varying the configuration of the composite. Disks are one of such configurations of the composite [33]. This paper presents a ring-type resonant magnetic field sensor. Figure 3 shows the schematic of a magnetic field sensor consisting of a PZT disc, but PMN-PT and a Terfenol-D magnetostrictive disc can also be used.

PZT discs were of dimensions of 15.25 × 0.85 mm^2^. The diameter of the inner dot was 5.04 mm, and that of the outer ring was 8.07 mm.

The presented bilayer DC magnetic field sensor operates on the basis of a ring-dot piezoelectric transformer made of a PZT plate and a glued Terfenol-D disc. The sensor detects the change in H_dc_ from the shift of the resonant frequency through the ME interaction when an electric voltage is applied to the inner electrode. The sensitivity of such a constant magnetic field sensor will depend on the ME voltage coefficients, the saturation magnetic field, and the electric field amplitude for the excitation of elastic nonlinearity. This sensor had a sensitivity of 10^−12^ T and a low noise level of 2 × 10^−11^ T/Hz^1/2^. In this case, the required bias field to obtain such values was 4 kOe, and the ME voltage coefficient was 0.12 V/(cm∙Oe). The studies carried out have shown the promise of using ring structures and set the task of reducing the saturation fields. The authors of [34] showed the possibility of decreasing the resonance frequency and increasing the resonance-enhanced ME coefficient of the ME structure by using an elastic-steel layer with a relatively high Q_m_. As for the usual structure Terfenol-D/PZT 15 mm long in LT mode, the resonance frequency was in the range of 80 kHz and the ME coefficient was approximately 18.5 V/(cm∙Oe) (Figure 4b); then, when using a three-phase ME structure on the basis of a steel layer and two PZT unimorphic plates in the bending mode (Figure 4a), the resonant frequency significantly decreased to *f*_res_ = 5.1 kHz and the magnitude of the ME coefficient increased to α_E_ = 40 V/(cm∙Oe).

In this paper, the maximum magnitude of the ME coefficient was observed at a bias field H_dc_ = 300 Oe, for both structures. A three-phase structure operating at bending resonance reached α_E_ = 40 V/cm∙Oe, and the unimorph of similar Terfenol-D and PZT layer geometries had a resonance frequency of 19.2 kHz and a resonance-enhanced α_E_ of 22 V/(cm∙Oe).

In 2016 [35], a giant ME effect was obtained in negative magnetostrictive/piezoelectric/positive magnetostrictive semiring structure (Figure 5). The research team managed to significantly reduce the bias field to 240 Oe, while maintaining the sensor sensitivity of 10^−12^ T and a low noise level of 2 × 10^−12^ T/Hz^1/2^, as well as to obtain a giant ME voltage coefficient for structures based on Terfenol-D 44.8 V cm^−1^∙Oe^−1^.

It was shown that the ME effect in the Ni/PZT/TbFe_2_ half-ring structure was much larger than in the TbFe_2_/PZT half-ring, which confirmed the different contributions to the ME coefficient of the components with negative (Ni) and positive (TbFe_2_) magnetostriction. It was found experimentally that with an increase in the radius of the half-ring, the resonance frequency decreased, and the ME coefficient increased. The giant ME voltage coefficient of 44.8 V∙cm^−1^∙Oe^−1^ was obtained for a radius of 13.0 mm, which made this design promising for a magnetic field sensor. Summarizing the above, one can see the development trajectory of magnetic field sensors based on Terfenol-D/PZT and Terfenol-D/PMN-PT. It can be seen that different research groups used separate types of composite structures (disc-shaped, rectangular, etc.). The most promising is a magnetic field sensor based on a negative magnetostrictive/piezoelectric/positive magnetostrictive semiring structure [35]. The magnitude of the ME coefficient in terms of voltage in this sensor reached values α_E_ = 44.8 V∙cm^−1^∙Oe^−1^, with relatively identical values of sensitivity and noise level. From the above, it can be seen that the magnitude of the bias field for each of the presented sensors varied in the range from 240 Oe to 4 kOe, which is the main disadvantage of structures based on Terfenol-D.

### 2.2. Metgas PZT/PMN-PT

Sensors based on Metglas-PZT are easy to manufacture and at the same time have good characteristics. Metglas provides a low bias field and a small hysteresis. PZT has a high piezoelectric coefficient, which results in a high magnetoelectric coefficient with a low bias field and good linearity of the sensor characteristics. Table 1 shows the main characteristics of the original materials. All source materials are widely distributed in the market and available.

It should be noted that the general aspiration to study the possibilities of the ME effect, its magnitude, and its features seems to have changed to a more pragmatic position of ME sensors manufacturing that are competitive with commercially available devices. Recently, a large number of works in this research field have been published, and interesting reviews on this topic have also been presented. Let us consider several works published in recent years that reflect the trends in the development of the science of the ME effect for practical applications of ME magnetic field sensors using ME layered composites of the PZT/Metglas and PMN-PT/Metglas.

An important parameter that determines the quality of the magnetic sensor is equivalent magnetic noise. The following papers prove the high parameters that are achieved by ME magnetic field sensors. In [18], Wang et al. investigated an equivalent magnetic noise spectrum for which the charge noise density spectrum with the ME charge coefficient was conversed. The structure of the ME sensor, made by the article authors, consists of six layers of magnetostrictive material Metglas and a core made of a piezoelectric composite containing of five PMN-PT piezofibers; there is also a pair of Kapton interdigited electrodes. The ME coefficient was measured and had a maximum value of 52 V/(cm·Oe) with a direct current magnetic bias of about 8 Oe at an alternating magnetic field frequency of 1 kHz. At a frequency of 1 Hz, the authors found an equivalent magnetic noise of 5.1 pT/√Hz, which is extremely low. Such a low equivalent magnetic noise, obtained on the studied ME structure, arose due to the low charge noise density in combination with the big ME charge coefficient. The ME ultra-low magnetic field sensors with the low equivalent magnetic noise can be especially effectively used for measuring the ultrasmall magnetic fields.

The sensor manufacturing technology and constructional materials are important, and the authors of many studies offer new solutions that are included in the knowledge base of research in this area. Jahns et al. in [36] conducted research of the noise level of ME sensor from thin-film material. The measurements were made at room temperature. In this study, a sensor containing a silicone cantilever with dimensions of a 20 × 3 mm^2^ and 625 μm thick was used. The molybdenum electrode with thick of 0.3 μm was covered to the Si cantilever from bottom, and a piezoelectric aluminum nitride with thick of 1.8 μm was used on top. The third upper layer was the FeCoBSi magnetostrictive electrode with a thickness of 1.75 μm, and area was A = 1.6 × 10^−5^ m^2^. The trench was etched on the back side of the Si-substrate on 7 mm wide. This made it possible to reduce the resonant frequency of the bending beam up to f_res_ = 330 Hz. The sensor was attached to the clamping unit with an epoxy adhesive. Using an optimal bias field of H_dc_ = 6 Oe and a sinusoidal field H_ac_ with f_ac_ = f_res_ = 330 Hz, the authors measured an ME coefficient of α_E_ = 1200 V/(cm∙Oe). The article presents the results of studies of the signal-to-noise ratio of the sensor in order to increase its sensitivity. The measurement results provided a sensitivity value of 5.4 pT/√Hz at a resonant frequency about 330 Hz.

Often, new technical solutions allow researchers to achieve competitive results. Wang et al. in [37] theoretically and experimentally researched the equivalent magnetic noise in the ME sensor. Was used three-layer Metglas (80 × 10 × 0.025 mm^3^)/0.7Pb(Mg_1/3_Nb_2/3_)O_3_–0.3PbTiO_3_ (PMN–PT single crystal layer (14 × 6 × 0.8 mm^3^ with <110> direction along thickness)/three-layer Metglas (80 × 10 × 0.025 mm^3^) laminate structure thickness polarized mode (L–T mode) and longitudinally magnetized. DC leakage resistance (NR) and noise sources of dielectric loss (NDE) was researched. The Metglas layers and PMN–PT layer were glued together with West System 105/206 epoxy glue. An electrically insulating film was used in the ME structure between the Metglas layers for to avoid electrical breakdown. In the low-frequency region (1 Hz), NR predominated 1.6 times > NDE, and noise was of 9.1 pT/√Hz on 1 Hz, according to theoretical calculations. The equivalent magnetic noise in experimental research was 10.8 pT/√Hz. Moreover, the optimal DC bias of H_dc_ about 6.2 Oe was found, in which the ME charge coefficient α_Q_ = 3000 pC/Oe, and maximum value α_E_ = 19.5 V/(cm∙Oe).

A thorough knowledge of the technology used is very important for researchers, and therefore it is important to consider in detail the technological aspects of manufacturing sensors. The performance of lamination process effect on the composite ME sensors of Metglas/Pb(Zr,Ti)O_3_ was researched by Li et al. in [38]. PZT fiber pack dimensions of 4 cm × 1 cm consisting of five fiber dimensions of 4 cm × 0.2 cm, which were installed on a tape with low stickiness to the coated electrode, was attached. Six 8 × 1 cm pieces of Vitrovac 7600F Metglas (Vitrovac 7600F; Vitrovac Inc., Hanau, Germany) were furnace annealed at 350 °C for 1 h. Then the pieces of Metglas were glued together with West System 105/206 epoxy glue. The use of Metglas with big magnetostriction (λ about 42 ppm), also cure of interphase epoxy and spin-coat/vacuum-bag process, was applied. It has led to an increase of the ME coefficient from 6 up to 21.6 V/(cm·Oe), and a decrease equivalent magnetic noise from 2 × 10^−10^ to 4 × 10^−11^ T/√Hz in the 1 Hz range.

Often, the authors use similar technological techniques and design; in this case, minor details, design features, or the accuracy of the measuring installation can be significant. Wang et al. [39] investigated the tunability of ME properties by applied DC electric field for ME composite (Metglas/Pb(Zr_x_Ti_1−x_)O_3_ (PZT-fiber). The ME structure consisted of six layers with the dimensions of 80 × 10 × 0.025 mm^3^ each of amorphous magnetostrictive material Metglas (Metglas Inc.) and a piezoelectric PZT five fibers composite with dimensions of 40 × 2 × 0.2 mm^3^ each (Smart materials Comp., Sarasota, FL, USA). The layers had a pair of Kapton^®^ interdigitated (ID) electrodes with a spacing of s = 1 mm. The dependence of ME coefficients and equivalent magnetic noise on electric field were researched. The noise charge density and dielectric properties were comparatively steady to change of electric field. An improvement by 1.3 times in the ME coefficient was obtained at the Villari point under electric field strength of 300 V/mm relative to 0 V/mm. The maximum ME coefficient α_E_ achieved of 25 V/(cm·Oe). The equivalent magnetic noise was measured for electric field strengths of 0, 300, and 600 V/mm were 20.1, 15.0, and 25.6 pT/Hz^1/2^, respectively, at a frequency of 1 Hz.

Sometimes, the experimenter needs to calculate the optimal conditions to obtain the desired effect in order to perform experiments. In [40], two methods to effectively self-stress induce on Metglas/Pb(Zr,Ti)O_3_/Metglas ME composite are considered: applying a DC electric field to the core piezoelectric composites and other applying a DC magnetic field to the layers of Metglas. Multi-push–pull L-L mode Metglas/PZT/Metglas composites were fabricated during the research. A bundle of 40 × 10 mm PZT, consisting of five fibers 40 mm × 2 mm PZT-5A (Smart Materials, Sarasota, FL, USA), was used as the piezoelectric core of the composite. Two interdigitated Kapton electrodes on the bottom and top surfaces were glued on the piezoelectric core with epoxy (Stycast 1264, Henkel Corporation, Rocky Hill, CT, USA). Vacuum-bag and spin-coat technologies were used to mount the interdigitated electrodes on PZT core composites. The piezoelectric material was poled with 20 kV/cm electric field. Three Metglas layers with dimensions 80 × 10 mm (Vitrovac 7600 F, Hanau, Germany) were installed on the both sides of the piezoelectric composites core. The ME effect in the laminates can be enhanced by optimum self-stress. The value of α_E_ for the self-stressed ME structure was 31.4 V/(cm·Oe), with a bias field of 20 Oe. The equivalent magnetic noise floor was reduced from 13.3 pT/√Hz to 9.8 pT/√Hz at a frequency of 1 Hz for the self-stressed laminate in the frequency range of 0.1 < *f* < 30 Hz, as shown in the measurement results.

The method of analogies or methods used in similar situations may well be used to increase the obtained parameters. An ultralow equivalent magnetic noise in a bimorph heterostructure sensor of 6.2 pT/√Hz at 1 Hz was obtained in [19]. The ME sensor consisting of Metglas layers and a transverse-poled single crystal of PMN-PT (1 mol % Mn-doped Pb(Mg_1/3_Nb_2/3_)O_3_–29PbTiO_3_) was grown from a melt by a modified Bridgman technique. As-grown single crystals were oriented along ‹001›, ‹011›, and ‹0–11› directions, then cut for preparing fibers with dimensions of 30 × 2 × 0.2 mm^3^, and the fibers were organized with their ‹001› and ‹011› crystallographic axes oriented in the thickness and length directions. The Metglas was purchased from a commercial company with next composition of Fe_74.4_Co_21.6_Si_0.5_B_3.3_Mn_0.1_C_0.1_ (Vacuumscheltze GmbH & Co. KG, Hanau, Germany). The layer thickness was 25 µm and its dimensions were 80 × 8 mm^2^. Then, 12 of Metglas layers were packed and glued with epoxy glue (West system 206, Bay City, MI, USA). A vacuum bag pressure method was used and then poled. Mn-doped PMN-PT plate was packed and joined with the Metglas layers with epoxy glue (Westsystem, Bay City, MI, USA) and a drip at its ends of Silver paint (Ted Pella, Inc., Redding, CA, USA). The measured equivalent magnetic noise was less than 1 pT/√Hz at a frequency of 10 Hz. The measured ME coefficient was about 61.5 V/(cm·Oe).

Material suppliers, processing modes, and connections are important in technological research, wherein there are no small details; then, repeatable characteristics with high parameters are likely to be obtained. In the work of [41], Junqi Gao et al. published a study of Metglas/Pb(Mg_1/3_Nb_2/3_)O_3_–PbTiO_3_ ME laminated composites in which they presented a quasi-static charge amplifier method to detect the extremely low frequency response. The composite ME material was made for the study. To do this, the researchers obtained Metglas foils from Vitrovac (Sekels GmbH, Ober-Mörlen, Germany) and PMN–PT fibers from Korea Ceramic Co., Ltd. (Wonmi-gu, Korea). A layer with dimensions of 4 × 1 cm was formed from five 200 mm thick thin piezoelectric fibers, which were located along their long axes. The epoxide polymer insulating film coating was applied on the top and bottom surfaces of the piezoelectric layer using epoxy (Stycast 1264, Henkel Corporation, Rocky Hill, CT, USA), where on the films, metal interdigitated (ID) electrodes were applied. A single magnetostrictive layer with dimension of 8 cm in length and 1 cm in width was formed from three Metglas foils that first bonded together. Further, two such single magnetostrictive layers were then laminated to both sides of the piezoelectric material surfaces in order reach to the optimum volume ratio. The value of the measured charge coefficient α_E_ was about 2 × 10^−5^ C/T and 28 V/(cm·Oe). Measurements of the ME effect of Metglas/PMN-PT ME composites at extremely low frequency were carried out. The effect measured over the frequency range from 0.01 Hz to 1 kHz ME showed good stability. As a result, the ME magnetic sensor at broader frequency bandwidth was designed. The measured equivalent magnetic noise density of the sensor at 10 mHz was 3 nT/√Hz.

It is the case that an intuitive approach at first glance produces a result, and then it is confirmed by theoretical calculations and practical research. In [42], Wang et al. reported about ME properties in a dumbbell-shaped metallic glass alloy/Pb(Zr_x_Ti_1−x_)O_3_ laminated composite. Such a composite, because of the result of magnetic flux concentration effects of the dumbbell-shaped design, yields a magnetic field amplification. The authors described the ME composite, which was formed of six layers of dumbbell-shaped magnetostrictive Metglas and a two PZT piezoelectric fibers with dimensions of 40 × 2 × 0.2 mm (Smart Material Corp., Sarasota, FL, USA). Moreover, the sensor structure contained a pair of Kapton (DuPont, Wilmington, DE, USA) interdigitated (ID) electrodes with width of 20 µm and the distance between adjacent electrodes of 0.5 mm attached to each surface of the two PZT fibers using an epoxy. The Metglas foils had thicknesses of 25 µm and widths of 18 mm (Vacuumschmelze GmbH & Co. KG, Hanau, Germany), annealed at 300 °C. Metglas dumbbell-shape had a total length of 80 mm and end-flange widths of 18 mm, with end-flange lengths of 10 mm. The center cavity dimensions were 40 × 4 mm. The authors set a goal to achieve maximum magnetic flux concentration and at the same time reduce the size for the next practical use of the sensor. The ordinary rectangular shape of Metglas had a relatively larger value of best bias (H_dc_ = 5 Oe) and a lower maximum α_E_ of 5.5 V/(cm·Oe). At the same time, the optimal H_dc_ for the 18 mm dumbbell-shaped sensor design was decreased to 3.5 Oe and the maximum value of α_E_ was increased to 8.5 V/(cm·Oe). In the conducted study, the authors measured the noise voltage if the intentional excitation was missing, therein exhibiting a voltage density of 6.6 × 10^−4^ V/√Hz at a frequency of 1 Hz. The modified geometry of the ME structure compared with the simple rectangular structures led to an efficient enhancement in the value of ME coefficient, a strong drop in the requisite magnetic bias field, and an improvement in magnetic field sensitivity.

New technologies often provide a good result. In [43], Nasrollahpour et al. presented a paper dedicated to a miniature complementary metal oxide semiconductor oscillator with the application microelectromechanical system that has a resonant frequency at 159 MHz. The oscillator circuit was simulated and designed in 0.35 µm XFAB technology. The quality factor of manufactured ME sensor was equal to 653. Phase noises of the researched oscillator equal to −131.3 dBc/Hz at frequency 10 kHz, and −137.9 dBc/Hz at 100 kHz offset frequencies also were found when power of 2.24 mW was consumed.

New effects often outstrip the understanding of their wide practical use in household appliances. In the work [44], Staruch et al. reported that the magnetic field sensitivity reached up to 8% *f*_0_/mT if it modulated in a bending mode stress tunable sensor by uniaxial tensile stress. The magnetic field sensors with symmetric and asymmetric design were manufactured from a piezoelectric fiber composite, a few layers of magnetostrictive material Metglas, and interdigitated electrodes. Minimum equivalent noises for the present sensor modes amounting to 428 and 20 nT/√Hz at a frequency of 10 Hz were found. The frequency shift detected using the phase-locked loop circuit allowed for the determination of the minimum magnetic noise floor. According to the study, this noise floor directly corresponds to the maximum sensitivity of the sensor to the magnetic field, which depends on the effect of ΔE. In the next paper, the ultimate characteristics of the equivalent magnetic noise level of the ME sensors that are achievable to date are discussed.

A detailed consideration of the theoretical and practical aspects of the object under study allows one to achieve the best result. Viehland et al. in paper [7] discussed extremely sensitive ME magnetic field sensors. Such ME sensors consist of magnetostrictive and piezoelectric phases and using ME bulk and thin-film structures. The samples include components such as amorphous magnetostrictive material and PZT fibers or AlN as the piezoelectric component, wherein they can be multilayered. Noise floors can depend on the size of the ME sensor and the working regime, and may depend on the operating mode, wherein this floor is usually within 1–100 pT/Hz^1/2^ at frequency 1 Hz. Note that in paper [41], Wang et al. observed what is according to the Villari effect of the amorphous Metglas alloys significant E-induced increase of the ME coefficient, as well as a decrease of the equivalent magnetic noise. The authors also found that the ME coefficient will increase, and the equivalent magnetic noise will decrease under the action of an optimal electric field strength of 300 V/mm.

One of the main parameters of the sensors is sensitivity. Huong Giang et al. in [45] in their studies of the ME vortex magnetic field sensors describe the work carried out on modeling and experiments, including with geometry optimization (Figure 6). The research was carried out with a change in the size of the Metglas/Piezoelectric (PZT) structures of both closed and open magnetic circuit geometries. The authors report that they implemented the ring shape ME double sandwich current sensor, wherein the structure contained four layers of Metglas in a closed magnetic circuit. The sensor dimensions were 6 mm in diameter with a ring width of 1.5 mm. At the sensor resonant frequency of 174.4 kHz, the maximum output signal was about 95 mV. The developed sensor had sensitivity of 5.426 V/A at the resonant frequency.

The characteristics of the studied structure suitable for the magnetic field sensor design are considered in the example of one of the papers. Nana Yang et al. in [46] report on the study of an ultra-sensitive ME heterostructural sensor consisting of Metglas/Pb(Zr_0.52_Ti_0.48_)O_3_ (PZT thick films with a thickness of 2.2 µm), the distinctive properties of which are flexibility and cost-effectivity. The piezoelectric material has the extremely great piezoelectric coefficient d_33_ of about 72 pC/N for PZT thick films. As a result of this design, the flexible sensor shows a big ME coefficient at low frequencies of 19.3 V/(cm·Oe) and 280.5 V/(cm·Oe) at resonant frequency. The flexible ME sensor shows a good mechanical endurance and also possesses good sensitivities of 200 nT at low frequencies and 200 pT at resonant frequency. The sensors were subjected to 5000 bending cycles (radii about 1 cm) and yet showed no fatigue-induced productivity degradation. The flexible ME structure was manufactured and consisted of several layers as shown in Figure 7. A six-layer Metglas pack was joined to the bottom side of the mica-based PZT film with gluing epoxy at room temperature (West system 206, USA). A vacuum bag pressure method was used. Each Metglas layer was made with a thickness of 25 µm and an area of 40 × 8 mm^2^. The 100 nm thick Pt interdigital electrodes (IDEs) were deposited onto PZT thick films by means of magnetron sputtering combined with a custom-designed mask. The mica substrate for increased flexibility was thinned to 10 µm by scotch tape. The ME coefficient α_E_ for flexible structure Metglas/PZT was measured by applying AC magnetic field of 0.1 Oe and bias magnetic field from 0 up to 30 Oe at a frequency of 1018 Hz. The coefficient α_E_ has a highest value about 19.5 V/(cm·Oe) at a one-layer Metglas under bias magnetic field 4.5 Oe.

Figure 8a shows the output voltage signal of the ME flexible Metglas/PZT structure under the action of AC magnetic field at a frequency of 1018 Hz and under bias field 4.5 Oe. The ME output voltage signal demonstrated linear response to AC magnetic field. The change of ME voltage could not be observed if the amplitude of AC magnetic field was smaller than 100 nT. Hence, the limit of detection (LoD) of the ME sensor was only 200 nT. As it is shown in Figure 8b, the output voltage of ME flexible sensor vs. AC magnetic field at resonant frequency under optimal bias field 4.5 Oe was measured. The obtained magnetic field sensitivity of the AC magnetic field 0.2 nT at resonance frequency was much more than that at 1018 Hz. The authors consider that the main advantage of the proposed flexible structure is that it is prospective for applications in biomedicine.

A detailed description of the work performed and the resulting characteristics are of great interest, allowing us to assess the acceptability of the technology used. In [47], Gao et al. introduced ME laminate sensors consisting of magnetostrictive and piezoelectric phases. They made a comparison of the ME response and magnetic field sensitivities of engineered ME sensors. The ME voltage coefficients for structures Metglas/Pb(Mg_1/3_Nb_2/3_)O_3_–PbTiO_3_ (PMN-PT single crystal fibers) and Metglas/Pb(Zn_1/3_Nb_2/3_)O_3_–PbTiO_3_ (PZN-PT single crystal fibers) was no less than 8.5 V/(cm·Oe), being about 2.8 times larger than for the structure Metglas/Pb(Zr,Ti)O_3_ (PZT ceramic). This measurement data showed that for the single crystal structures, the sensitivity to magnetic field increased 1.7 times. The noise floors were about three to four times lower for PMN-PT or PZN-PT fibers than those with composites with PZT. The following materials were used in the work: Metglas (Metglas Inc., Anderson, SC, USA), PZN-PT single crystals (Microfine Materials Technologies Pte Ltd., Singapore), PMN-PT single crystals (Shanghai Institute of Ceramics, Shanghai, China), and PZT (CTS, Albuquerque, NM, USA). Then, 200 µm thick piezoelectric fibers were cut to the dimensions of 2.5 × 0.4 cm^2^, and one and the other surfaces of the fibers were adhered using an epoxy glue to thin polymer insulating films with IDEs. The piezoelectric fibers were arranged to symmetrically pole in a back-to-back pattern along their length axis due to electrode design. Piezo-structures were laminated together using an epoxy between four layers of Metglas with 25 µm thickness and size of 8 × 0.4 cm^2^. PZN-PT- and PMN-PT-based ME sensors have sensitivity to magnetic field at 0.6 nT, which were about 1.7 times larger than for PZT-based ME sensor with only 1 nT sensitivity. Metglas/PMN-PT- and Metglas/PZN-PT-based ME sensors had the noise floors of 60 pT/√Hz in the frequency range 10 up to 100 Hz and of 20 pT/√Hz in the frequency range 0.2 up to 1 kHz. The noise floors in the low frequency range for the Metglas/PZT-based sensor was 150 pT/√Hz, and in the high frequency range slightly higher at 70 pT/√Hz.

The value of the bias magnetic field for the ME sensor, in addition to noise and sensitivity, is an important parameter. An electrically driven bulk magnetic field sensor built on the converse ME effect was studied by Chu et al. in [48]. Figure 9 schematically shows the ME structure inside a pick-up coil. The crystal Pb(Mg_1/3_-Nb_2/3_)O_3_-Pb(Zr,Ti)O_3_ (thickness-poled PMN-PZT [011]-oriented, Ceracomp Co., Ltd., Cheonan, Korea) with dimensions of 30 × 1 × 0.2 mm was used as ME laminate core. Five layer of Metglas fiber with dimensions of 52 × 1.5 × 0.125 mm were located on the bottom and top side of the ME laminate core. ME laminate of 1-1 type from the piezomagnetic and piezoelectric fibers was finally formed by bonding together with epoxy resin. The authors experimentally established that, without any magnetic bias field, exciting the ME laminate at 1 V achieved a LoD about 115 pT for a magnetic field at the frequency of 10 Hz and about 300 pT for a magnetic field at the frequency of 1 Hz. The achieved power consumption for such ME structure was 0.56 mW.

In paper [49], Dong et al. demonstrated an extremely sensitive ME sensor based on Metglas/PZT fiber heterostructures. Such a sensor operates in a completely demagnetized state of Metglas. PZT/Metglas ME sensors have shown magnetic noise level of 9.1 pT/√Hz in the magnetic field detection at the frequency 1 Hz and LoD below 2 pT. ME composites have achieved sensing of ultra-low magnetic field below 1 pT and also outstanding ME conversion coefficient. Moreover, earlier, Chu et al. in paper [50] described 1D (1-1) connectivity ME composites consisting of a [011]-oriented PMN-PT single-crystal fiber that was laminated together with laser-treated Metglas. In L-T mode, resonant ME coupling coefficient reached about 7000 V/(cm·Oe). In this work, the magnetic sensitivity of 1.35 × 10^−13^ T at resonance frequency and at room temperature was achieved. This is a significant achievement that allows us to declare the hypersensitivity of the ME magnetic field sensors.

In this review, we have also included some interesting works that provide an idea of the level of development of research in the field of ME sensors. In [51], Chu et al. proposed a design of 1D magnetic sensor array consisting of 56 ME sensors for implementation of a magnetic detecting and sketching system. The ME sensor unit was manufactured from a (1-1) connectivity Metglas/PMN-PT composite operating in the L-T mode. PMN-PT fiber with dimensions of 30 × 1 × 0.2 mm (thickness-poled [011]-oriented, Ceracomp Co., Ltd., Cheonan, Korea) was used as a piezoelectric phase with high transverse piezoelectric coefficient d_32_ equal to 1850 pC/N. Five layers of Metglas fiber with dimensions of 100 × 1.5 × 0.125 mm was used as a piezomagnetic phase with highest permeability. Then, both phases were glued together with epoxy. The (1-1) ME sensor had the ME coupling coefficient α_E_ at its resonant frequency at about 22 kHz, reaching to 28 V/(cm·Oe) at a frequency of 1 kHz and 5600 V/(cm·Oe) under the optimal DC magnetic bias H_DC_ = 2 Oe. In [27], Li et al. studied the properties of the Metglas/PZT magnetoelectric composite, particularly the nonlinear ME effect at the frequency of electric-mechanical resonance of 30 kHz. For the manufacturing of Metglas/PZT composite, they used five PZT fibers (oriented along the length direction of the laminate, Smart Materials) with dimensions of 40 mm × 2 mm × 180 μm. Two Kapton films with IDEs were glued by epoxy resin to both surfaces of the PZT fibers. Four layers with dimensions of 80 × 10 mm of MG foils (Vitrovac 7600F) were bonded on both sides of the PZT piezoelectric layer. The measured linear ME coefficient α_E_ was equal to 23.2 V/(cm·Oe) at the maximum when the bias field was 10 Oe. The sensitivity of the nonlinear ME sensor to DC magnetic field was 17.5 V/Oe, which allowed for the detection of the DC field at about 2 × 10^−5^ Oe. Leung et al. in the report [52] studied ME devices based on Metglas/PZT composite, as well as the magnetostatically tunability of magneto-impedance and magneto-capacitance devices. Four kits of L-T with three layers of Metglas/PZT structure were manufactured. Metglas was annealed at different temperatures. The PZT plates (APC Piezo, Mackeyville, PA, USA) were cut into sizes of 30 × 5 × 0.5 mm, and the Metglas foils (Vacuumscheltze GmbH & Co. KG, Hanau, Germany) had a thickness of 25 µm and were cut into sizes of 28 × 5 mm. Annealing of 1 layer and 10 layers of different Metglas foils was carried out at temperatures of 350, 400, 450, and 500 °C for 30 min. ME structures were stacked together and glued with an epoxy resin, then cured for more than 4 h at 60 °C using the vacuum bag method. The annealing temperature of Metglas influenced the degree of tunability, as it was found. The sample with Metglas layers annealed at 500 °C showed at the EMR frequency an impedance tunability more than 400%, and at 350 °C, tunability was a factor of two higher. As it has been researched, the tunability of the capacitance at annealing temperature of 500 °C was found to be 290% at resonance frequency and 135% at antiresonance frequency.

The reviewed articles show the progress made in the research of ME magnetic field sensors. In general, the sensitivity of such sensors shows the ability to measure ultra-small magnetic fields with a high degree of precision. As can be seen from the considered theoretical and practical studies, the sensitivity of promising magnetic field sensors made on the basis of composite layered materials Metglas/PZT and Metglas/PMN-PT can be less than 2 nT. Equivalent magnetic noise in the designs of such sensors is approximately no more than 5 ÷ 10 pT/√Hz at 1 Hz. Ultra-sensitive ME sensors can have the greatest potential in applications for measuring weak biological fields, remote sensing, and designing MEMS devices.

### 2.3. Metglas—Lithium Niobate

Ferroelectric lead-free crystals have a great potential to be used in ME composite structures because of their high-quality factor, stable electromechanical properties, and comparable values of d/ε with composite structures based on PZT and PMN-PT [53,54]. Reference [55] shows different ME structures based on y +128° lithium niobate (LN): unimorph (single domain), bimorph (two single domain LN crystals bonded to each other with epoxy adhesive), and bidomain (crystal with bimorph-like structure made by diffusion heating technology [56,57,58,59]). A layer of Metglas (2826 MB) was attached to the structure by epoxy adhesive for creating ME structure. Linear dimensions of all samples were equal: LN plate—10 × 10 × 0.5 mm^3^, Metglas layer—10 × 10 × 0.029 mm^3^. Measurements of ME coefficient α_E32_ were conducted by two methods in order to compare ME structures: by quasistatic method (AC magnetic field applied at frequency far from resonance *f* = 1 kHz with constant amplitude δH = 1 Oe and simultaneously varied DC magnetic field) and by dynamic method (AC magnetic field varied over wide frequency range with constant amplitude δH = 1 Oe, and simultaneously applied DC magnetic field value matches to the field, at which α_E32_ in quasistatic method obtained its maximum value for the investigated structure).

All structures showed significant growth of α_E32_ on resonant frequency because of electromechanical resonance (EMR) phenomenon. Maximum values of α_E32_ were obtained with bidomain “tail-to-tail” LN structure. α_E32_ was equal to 1.88 V/(cm∙Oe) for DC field H = 16 Oe measured by quasistatic method and to 462.7 V/(cm∙Oe) measured by dynamic method on resonant frequency f = 30.4 kHz. Dependence graph of equivalent magnetic noise density (EMND) from frequency is presented in Figure 10.

Experimental data had a good correlation with theoretical calculation. EMND was equal to 397 fT/Hz^1/2^ on resonant frequency.

According to the results, the most perspective structures were bidomain because bimorph structures contain epoxy adhesive in interphase between two crystals that decrease α_E32_, whereas unimorph structures have a minor increase of α_E32_ at bending resonant frequency.

The obtained results can be explained by intrinsic field appearing in crystal. A scheme of this process is presented in Figure 11a,b. In unimorph case, when the magnetic field is applied, the upper side of the crystal expands and creates an electric field that is parallel with crystal polarization while the lower side contracts and creates an electric field is antiparallel with crystal polarization. These fields exclude each other so that the final output signal is lowered. When it comes to ME based on bidomain crystals, vectors of created electrical fields will be equal, and thus output signal in bimorphs and bidomains is larger than in unimorphs. Moreover, there appears to be a bending mode EMR in bidomains and bimorphs. Output signal is much larger at resonant frequency, which is in the low-frequency range. This phenomenon can be used for low-frequency magnetic field sensors.

Record rate of EMND for ME sensors was reported in [60]. Investigated structure—bidomain LN y +140°/Metglas (2826 MB) bonded with epoxy adhesive. Choice of y +140° was related to LN crystals’ anisotropy, and value of PE coefficient d_23_ was at maximum in this direction.

Value of ME coefficient α_E31_ was 1704 V/(cm∙Oe) at resonant frequency 6862 Hz, and at non-resonant frequency (110 Hz) 1.9 V/(cm∙Oe). Measurement of EMND was in the frequency range from 10 Hz to 10 kHz. Maximum of sensibility of ME structure was obtained at a resonant frequency of 6862 Hz and was equal to 91 fT/Hz^1/2^ (Figure 12).

Results demonstrate the potential of ME bidomain LN structures to be used in the creation of supersensitive magnetic field sensors for biomedicine.

Reference [54] presents a comparison of ME structures with different lengths of structures and methods of fixing: cantilever (one end of sample is hard fixed) or free (sample is fixed between two probes in the middle of structure). When resonant properties are taken into account, samples with free fixing show better results than those with cantilever fixing. Maximum of α_31_ was equal to 478 V/(cm∙Oe) for a 45 mm sample with extrinsic DC magnetic field H = 2 Oe and at a resonant frequency 1335 Hz, and best sensitivity to magnetic field was obtained with a 30 mm sample with a value of 0.4 pT/Hz^1/2^ with H_DC_ = 6.5 Oe at *f*_r_ = 3166 Hz. However, for a low-frequency range, the best values of α_E31_ were obtained with cantilever fixing. Maximum of α_E31_ was obtained with a 45 mm sample at *f* = 10 Hz with a value of 7.2 V/(cm∙Oe). Magnetic sensivity was 126 pT/Hz^1/2^ with H_DC_ = 2 Oe.

Cantilever fixing structures have a lower bending mode resonant frequency in comparison with free fixing ones because they are more suitable for implementation in biomedical sensors.

Noise from acoustic vibrations dramatically decreases sensitivity of ME structures; thus, in [61], a new tuning fork construction is shown for dependence decreasing of noise signal.

Appearance and layer structure of construction is presented in Figure 13. The investigated sample was a composite structure of bidomain LN/Metglas in form of a tuning fork. One end of the sample is cantilever hard fixed. The first tine is covered with a Ta electrode on the upper side and with Metglas on the lower side. The second tine is made in the opposite way. When the AC magnetic field is applied, tines bend in opposite directions, while vibrations make them bend in the same direction. With cross-connection (Metglas layers are connected to each other, while Ta electrodes are also connected to each other), ME signals from tines sum up, while piezoelectric noise signals cancel each other.

Quasistatic ME coefficients’ values of tuning fork construction and each prong were 1.25, 1.3, and 1 V/(cm∙Oe), while in resonance, these coefficients were equal to 144.4, 150, and 144.1 V/(cm∙Oe).

In experiment ME, the tuning fork sample at resonant frequency 318.2 Hz showed a fivefold decrease of detected vibration noise in comparison with each tine. Moreover, there was a 7–25-fold decrease of vibration noise out of the resonant frequency, and thus an opportunity for non-resonant range was investigated to obtain better results. Figure 14 shows the comparison of EMND and LoD for the tuning fork and single-plate ME sensors.

At the resonance frequency, the EMNDs amounted to 3 pT/Hz^1/2^ and 20 pT/Hz^1/2^ for the tuning fork and single plate sensors, respectively.

It is shown that bidomain crystals have a greater ME effect compared to bimorph and monodomain crystals. The material parameters of bidomain crystals and magnetostrictive material is shown in Table 2. All ME sensors are sensitive to vibration noises, and thus an optimal structure for bidomain LN ME sensors is a tuning fork structure. The next steps of ME effect increase are the decreasing of the linear dimensions of the structure and the increasing of Metglas layer thickness.

### 2.4. Other Structures

Although most of the studies were carried out on structures such as Terfenol-PZT/PMN-PT, Metglas-PZT/PMN-PT, and Metglas-Lithium Niobate, in a number of works, new approaches were used and high characteristics of sensors were obtained. In contrast to the standard layered magnetostrictive-piezoelectric structure, Xing et al. [20] proposed a new composite based on a PZT bimorph and an NdFeB magnet. The new sensor showed a fairly high sensitivity of 500 pT at 10 Hz and acceptable linearity in the measurement range from nano to milli Tesla. In order to carry out biomagnetic measurements and replace bulky magnetic sensors of the SQUIDs type, Marauska et al. in [21] proposed the use of MEMS technology to create a miniature ME magnetic field sensor. When using a rectangular microcantilever with a thickness of 4 µm and dimensions of 0.2 mm × 1.12 mm, the authors obtained an ME voltage coefficient of α_E_ = 1000 V/A at a bending resonance frequency of 2.4 kHz. The calculated static ME voltage coefficient was α_E_ = 14 V/A. In the bending resonance mode, the following parameter values were achieved: the magnetic sensitivity of 780 V/T and the equivalent noise level of 100 pT/Hz^−1/2^.

Nan et al. in [23] proposed an ME sensor using the NANO-MEMS technology. A high-resistance Si wafer was used as a substrate, and AlN (250 nm thick) and FeGaB (250 nm thick) were used as piezoelectric and piezomagnetic layers, respectively. The self-biased sensor operated on the basis of the delta-E effect in the field of EMR at a frequency of 215 MHz and showed a sensitivity of 300 pT. The results obtained indicate that the proposed sensor can be successfully applied for technical and biomagnetic measurements, since it has a small size, compatibility with CMOS technology, and a relatively high sensitivity. The authors plan to increase the sensitivity of the sensor in the transition to the GHz range. In [26], Fetisov et al. demonstrated an ME sensor of an alternating magnetic field based on a nonlinear ME effect caused by the mixing of the measured field and the pump field due to the nonlinearity of the magnetostrictive layer. The structure of the ME element consisted of a plate of monocrystalline piezoelectric langatate (thickness 0.47 mm) glued to a Metglas tape (thickness 25 μm). The sensitivity measurements were carried out in the frequency range of 20 Hz–100 kHz, while the amplitude of the measured signal field reached 2 Oe, and the pump field reached 10 Oe. The measured maximum sensor sensitivity was 1.8 V/Oe, which was 320 times higher than the sensor sensitivity on a linear ME effect, but 4 times lower than the sensitivity of the linear sensor at resonance. The measurement limit of the magnetic field of the self-biased sensor was 10^–5^ Oe in the frequency range of 1–70 kHz, with a frequency resolution of 50 Hz. The tuning of the operating frequency of the sensor can be carried out by changing the dimensions of the structure and choosing the vibration mode. The sensor sensitivity depended on the pump field amplitude. Higher magnetic sensitivity in the mode of nonlinear ME effect of 17.6 V/Oe with a zero bias field was observed by Li et al. in [27]. They investigated the sensor with the PZT-Metglas composition at a frequency of 30 kHz (third harmonic) in the range of magnetic fields: the AC magnetic field from 0.1 to 10 Oe and a DC magnetic field from 0 to 20 Oe. The LoD of this sensor was of 2 × 10^−5^ Oe for the DC magnetic field, which was improved threefold compared to the ME sensor using linear ME effect. Annapureddia et al. in [28] described the self-biased ME sensors of an alternating magnetic field based on a PMN-PT single crystal and a flexible nickel plate. The authors, on the basis of the study of three ME sensors with variable sensitivity, showed that the main contribution to the magnetic sensitivity of the sensor is made by the piezoelectric component. Thus, using a low loss microfiber PMN-PT single crystal, they were able to obtain a magnetic sensitivity of 1 pT/√Hz at a resonant frequency of 500 Hz, as shown in Figure 15. Note that this value was obtained under ambient conditions and was much higher than that which was offered by commercially available magnetic sensors based on the giant magnetoresistance, fluxgate, or Hall effect.

As shown by the latest studies presented in [7] by Viehland et al., with electromechanical resonance, the LoD of ME sensors in the fT/Hz^1/2^ range can be achieved. At low frequencies, to increase the sensitivity, one is recommended to use frequency conversion using external alternating fields set frequency. Another way to increase the magnetic sensitivity of the sensor in the low frequency range is to use a change in elastic constants in an external magnetic field, i.e., delta-E effect. In this method, it is possible to obtain a limiting sensitivity of about 1–100 pT/Hz^1/2^ at a frequency of *f* = 1 Hz, taking into account the choice of the sensor design and its mode of operation.

Li et al. in [24] presented a highly sensitive DC/low frequency magnetic field sensor, using the NANO-MEMS technology, and applied a new measurement method based on the delta-E effect. The sensor design consisted of a layered structure (Figure 16) based on piezoelectric (AlN) and magnetostrictive (FeGaB) components.

To reduce eddy current losses, the authors placed layers of aluminum oxide between the FeGaB layers.

Unlike previous measurement circuits, where an induced voltage or impedance change was measured, the authors developed a new system that can measure the reflected electrical signal with a frequency shift due to the delta-E effect. The developed sensor showed a resonance frequency shift of 3.19 MHz (1.44%), which led to a high DC magnetic sensitivity of 2.8 Hz/nT, as shown in Figure 17.

At the same time, a LoD was 800 pT in an unshielded, room temperature and pressure, lab environment.

## 3. Discussion

One of the most important characteristics of an ME magnetic field sensor is its equivalent magnetic noise dependence on frequency. The magnitude of the sensor equivalent magnetic noise almost completely determines the main characteristic of the magnetic field sensor—the minimum measurable value of the magnetic field.

Below, we mark the places in the article where the noise characteristic of the sensor is discussed. In Section 2.2, Figure 8 shows the dependences for the low-frequency and resonant cases of the output voltage on the amplitude of the alternating magnetic field for the flexible ME sensor PZT/Metglas. In Section 2.3, the Figure 10, Figure 12, and Figure 14a show the frequency dependences of the equivalent magnetic noise for various ME sensors based on Metglas and lithium niobate. Figure 14b shows the dependences of output voltage for the tuning fork and single-plate ME sensors at resonant regimes on the amplitude of the alternating magnetic field. In Section 2.4, Figure 15 shows the dependences of output voltage for the Ni/PMN-PZT ME sensor at low-frequency and resonant regimes on the amplitude of the alternating magnetic field. The corresponding achieved minimum limits for measuring the magnetic field are shown. Table 3 summarizes the characteristics of the magnetic field sensors mentioned earlier in the article. Several types of physical quantities characterizing the sensitivity of ME magnetic sensors are given. Thus, the equivalent magnetic noise density characterizes the spectral density of the total sensor noise, reduced to the input magnetic field induction. The ME voltage coefficient is the ratio of the output amplitude of the electric field strength to the input amplitude of the magnetic field strength as a function of frequency. Knowing the ME coefficient for the voltage and the thickness of the ME structure, one can find the ME sensitivity of the magnetic field sensor. Sensitivity indicates how much electrical voltage the sensor generates when exposed to a magnetic field once. LoD is the minimum value of the magnetic flux density that the sensor can still measure. Moreover, one of the important parameters of the ME magnetic field sensor is the optimal bias field. This indirectly indicates the required parameters of the permanent magnet included in the sensor. This, in turn, has a rather strong effect on the mass and size characteristics of the sensor. Recently, in the development of ME magnetic field sensors, there has been a tendency to use a self-biased two-layer magnetostrictive structure, which makes it possible to abandon the permanent magnet. In order not to overload the table, we have not provided the optimal offset fields.

Comparison of the data in this table shows that in the low-frequency region, the most sensitive of the presented ME magnetic field sensors are those described in [46,47,48,49,62], and in resonance mode is a sensor based on bidomain LN y +140°/Metglas 2826 MB [60] and the sensor described in [50]. In practice, the most important characteristic of ME magnetic field sensors is the equivalent magnetic noise. Reducing the equivalent magnetic noise allows for the sensitivity of such sensors to be increased. Moreover, for alternating magnetic field sensors, the frequency range in which they can measure the magnetic field is of great importance. In some cases, it is necessary to measure the amplitude of a very low frequency alternating magnetic field. For example, for the use of ME magnetic field sensors in magnetoencephalography, the frequency band in which these sensors are sufficiently sensitive should be from 10 mHz to 1 kHz [63]. In this case, the main of studies of ME sensors of an alternating magnetic field were carried out in the frequency range from 1 Hz to 1 kHz [18,19]. To study such low frequencies from 10 mHz to 1 Hz, researchers should use more suitable ME structures [41]. Moreover, for these purposes, the noise of the amplifier must be reduced. To do this, one can use high-rated feedback resistors. However, making resistors with such large resistances with very small tolerances is very difficult. To increase the sensitivity of ME magnetic field sensors, a decrease in external noise is of great importance. One of the best opportunities for this is a gradiometry. In this method, several ME magnetic field sensors are combined in such a way that the signals of individual sensors are subtracted from each other in the total signal. This leads to the fact that the coherent external noise is compensated [64]. Gradiometer systems successfully combat coherent external noise but do little to combat the internal inconsistent noise of individual sensors. Moreover, the usefulness of the gradiometer method is reduced with serious differences between the individual ME sensors of the magnetic field of the system, especially when it comes to phase characteristics. Therefore, it is clear that for the successful application of the gradiometer method in ME magnetic field sensors, it is necessary to reduce the equivalent magnetic noise and the phase shift of the ME sensors that make up the system.

Theoretical and experimental studies of the nonlinear ME effect have shown that the modulation–demodulation method can be used to increase the sensitivity at an arbitrary frequency of the alternating magnetic field [15,62]. For the successful application of this method, the ME effect must have a narrow high maximum at the resonant frequency [15]. Therefore, it is necessary to study and design ME magnetostrictive-piezoelectric structures with a high mechanical quality factor. Thus, composite components with high mechanical figure of merit must be investigated and developed.

Important and promising types of ME magnetic field sensors are ME sensors based on the delta-E effect, that is, a change in elastic properties caused by a magnetic field. The delta-E effect [65] manifests itself in the relationship between deformation and magnetostriction, which leads to a magnetoelastic change in the direction of magnetization in unsaturated magnetic materials. The addition of magnetostrictive deformation to elastic deformation leads to a clear decrease in elastic moduli, the dependence on the magnetic field of which is used to measure the magnetic field.

The current operating modes are based on the detection of resonant detuning, induced by a magnetic field, in electromechanical bulk [23] and cantilever resonators [22]. Electrical sensing as well as mechanical excitation is achieved through the piezoelectric layer. The operating points are determined by the mechanical resonance frequency of the system *f*_r_ and the magnetic operating point, which is selected near the maximum change in the electromechanical resonance frequency as a function of the magnetic field. In this case, the magnetic field is converted into a frequency shift of the resonance. By simultaneously working with several mechanical modes and detecting them [66], the sensitivity of sensors based on the delta-E effect can be increased. An alternative approach is based on the dispersion of the magnetic field of the propagating phase shift of acoustic surface shear waves in the delay line sensor [67]. In general, delta-E sensors offer high performance broadband sensors suitable for detecting magnetic signals from heart and brain.

## 4. Conclusions

In conclusion, let us discuss the possible applications of ME sensors of the magnetic field and the problems that should be solved in the future. First of all, these are measurements of weak magnetic fields in various technical applications, such as in the search for iron ore deposits or in control systems during the movement of large magnetic masses, i.e., in cases where high magnetic sensitivity and mobility in control are required. The next important area of application is biomedicine, for which, in addition to the above-mentioned requirements, the following are also necessary: small dimensions of the sensor and its operation in a wide low-frequency range from 10 MHz to 1 kHz. In [25], data are given for the first successful application of a thin-film ME sensor in the bending mode in biomedicine for magnetoencelographic measurements. Taking into account the requirements for the frequency range and dimensions, we should consider ME sensors based on the delta-E effect to be more promising for biomagnetic measurements. In [22,29,68], such sensors with high sensitivity and micro-design are described, which can find wide application in biomedicine.

At the same time, for a wider application of ME magnetic field sensors, it is necessary, first, to increase the magnetic sensitivity and bring it to a level exceeding 700 fT/Hz^1/2^, in order to replace optically pumped atomic magnetometers [69] and giant magnetoimpedance sensors [70]. As follows from the analysis of the data in Table 3, such parameters can be achieved in the near future.

The second problem of ME sensors is the need to reduce the level of external noise. The use for this purpose of a system of two sensors or a gradiometer [64] made it possible, as a result of antiphase addition of individual noise signals, to sharply reduce the level of external noise and ensure the operation of the sensor at room temperature without additional shielding.

As a result, it can be noted that ME magnetic field sensors can provide passive operation at room temperature without additional shielding and are most effective for measuring weak magnetic fields in the low-frequency range. With their high magnetic sensitivity and low noise level at micro- and nano-sizes, the sensors are especially promising for use in biomedicine.

## Figures and Tables

**Figure 1 sensors-21-06232-f001:**
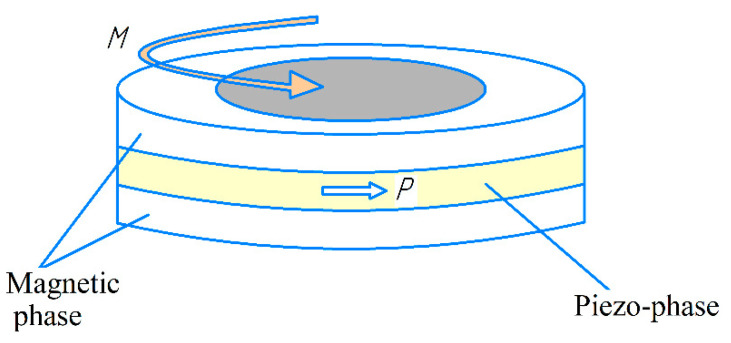
C-C mode of ring ME laminate: magnetic phase—Terfenol-D, piezo phase—PZT. Reprinted with permission from [31]. Copyright 2004 AIP Publishing.

**Figure 2 sensors-21-06232-f002:**
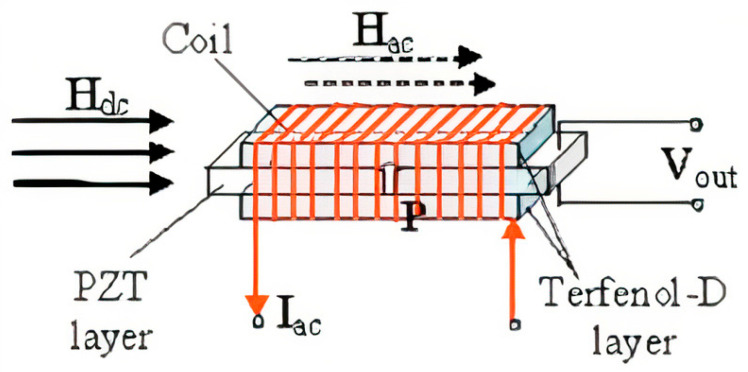
Illustration of DC magnetic sensor, which is an L-T mode trilayer laminate of Terfenol-D/PZT/Terfenol-D. Reprinted with permission from [32]. Copyright 2006 AIP Publishing.

**Figure 3 sensors-21-06232-f003:**
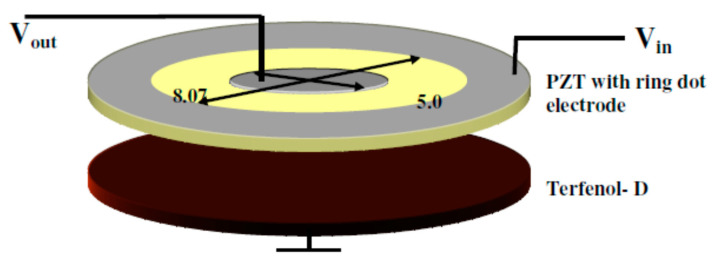
Schematic diagram of the DC magnetic field sensor. Reprinted with permission from [33]. Copyright 2008 Wiley.

**Figure 4 sensors-21-06232-f004:**
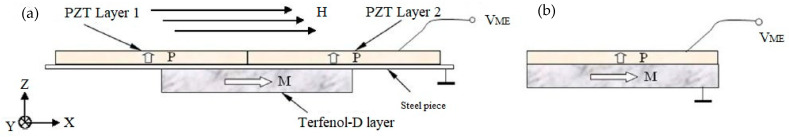
Illustration of ME laminate configurations (**a**) Terfenol-D/steel/PZT three-phase bending-mode unimorph and (**b**) Terfenol-D/PZT unimorph. Reprinted with permission from [34]. Copyright 2006 AIP Publishing.

**Figure 5 sensors-21-06232-f005:**
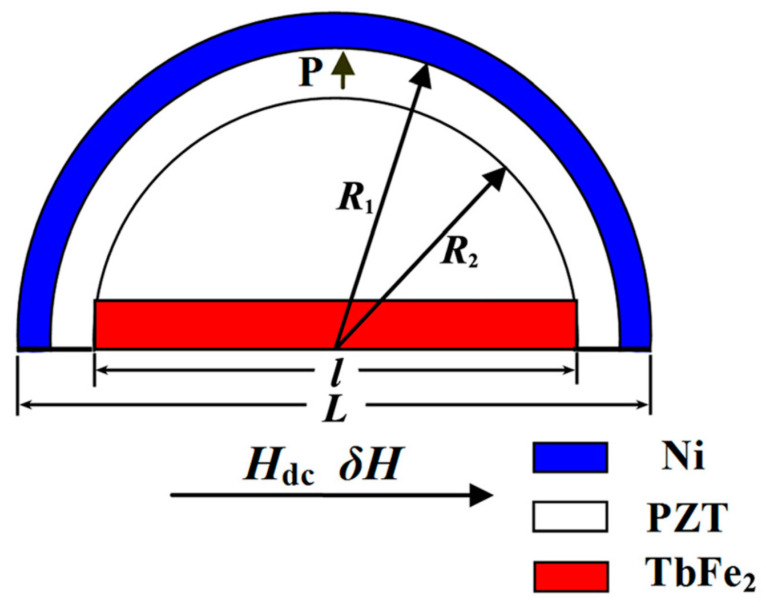
Schematic geometry arrangement of the Ni/PZT/TbFe_2_ semiring structure. Reprinted with permission from [35]. Copyright 2016 AIP Publishing.

**Figure 6 sensors-21-06232-f006:**
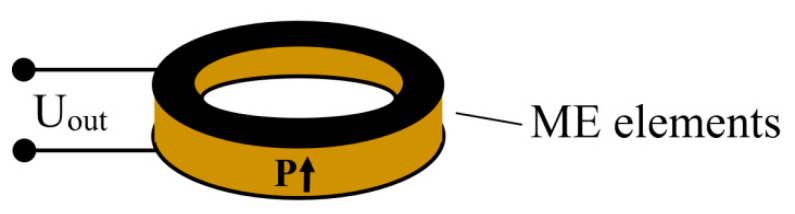
The schematic illustration of O-type ME structures. Reprinted with permission from [45].

**Figure 7 sensors-21-06232-f007:**
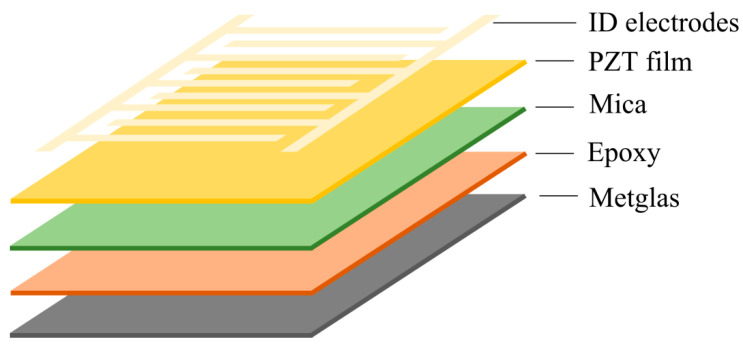
Flexible ME laminate structure. Reprinted with permission from [46]. Copyright 2012 AIP Publishing.

**Figure 8 sensors-21-06232-f008:**
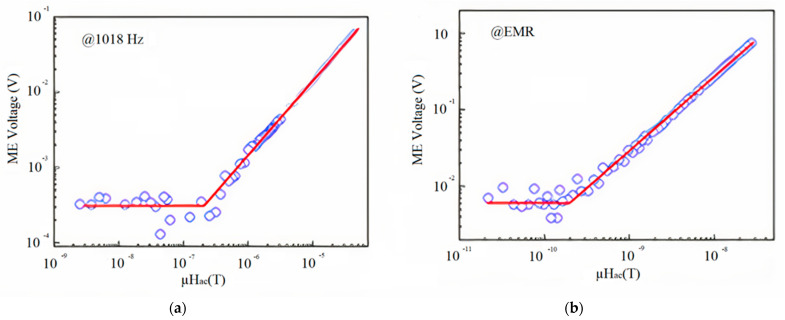
Magnetic field sensitivity and linear response of the flexible Metglas/PZT laminated ME sensor to the AC magnetic field at bias field of 4.5 Oe (**a**) at the frequency of 1018 Hz and (**b**) at the resonant frequency of 54 kHz. Reprinted with permission from [46].

**Figure 9 sensors-21-06232-f009:**
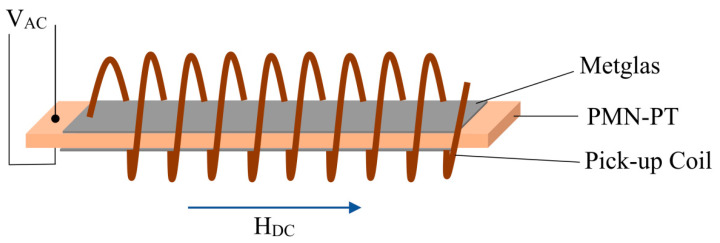
The ME laminate wound with a pick-up coil. Reprinted with permission from [48]. Copyright 2019 AIP Publishing.

**Figure 10 sensors-21-06232-f010:**
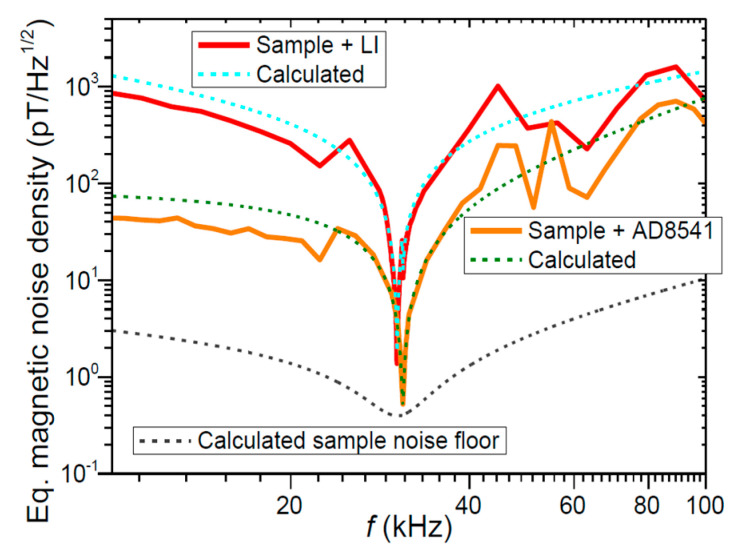
Experimental and calculated equivalent magnetic noise spectral density of the samples and detection circuit (lock-in and AD8541). Reprinted with permission from [55]. Copyright 2017 IEEE.

**Figure 11 sensors-21-06232-f011:**
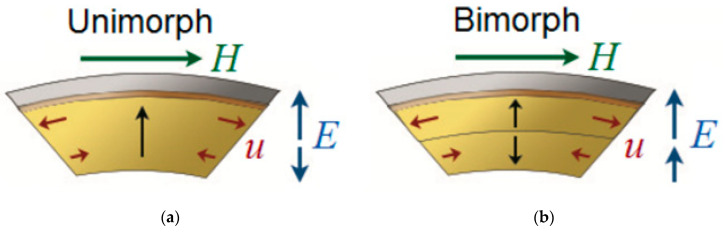
Scheme of intrinsic magnetic field creation in unimorphs (**a**) and in bimorphs (**b**). Reprinted with permission from [55]. Copyright 2017 IEEE. H—extrinsic magnetic field, u—crystal deformations, E—created intrinsic magnetic field.

**Figure 12 sensors-21-06232-f012:**
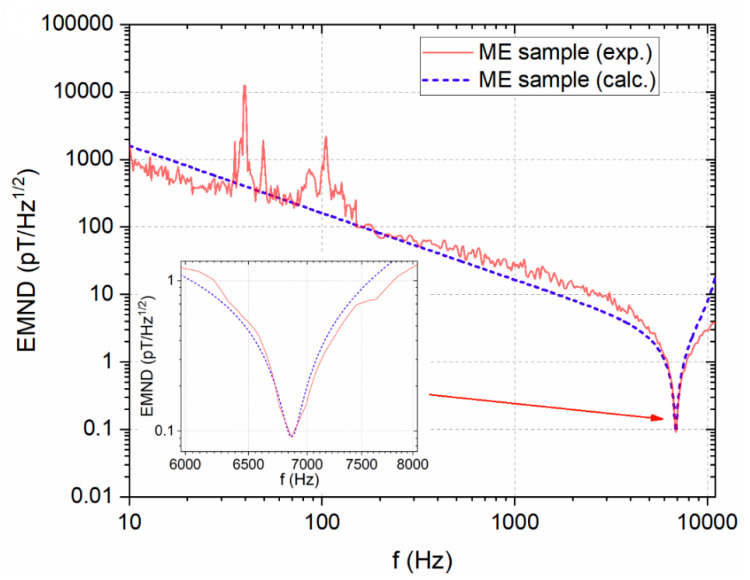
EMND measured as a function of the frequency in the Metglas/y + 140°-cut LN composite, as well as a calculated curve. Reprinted with permission from [60]. Copyright 2018 AIP Publishing.

**Figure 13 sensors-21-06232-f013:**
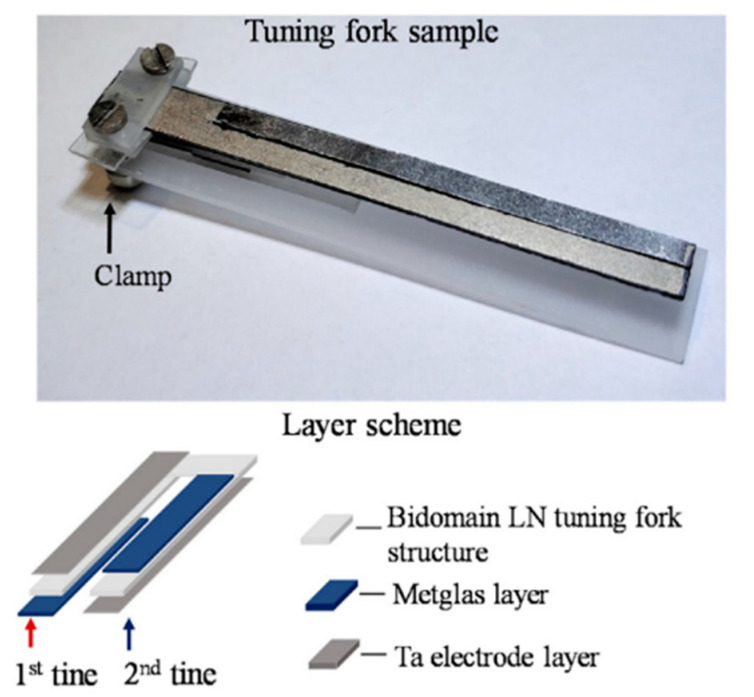
Appearance and layer structure of sensor construction. Reprinted with permission from [61]. Copyright 2019 Elsevier.

**Figure 14 sensors-21-06232-f014:**
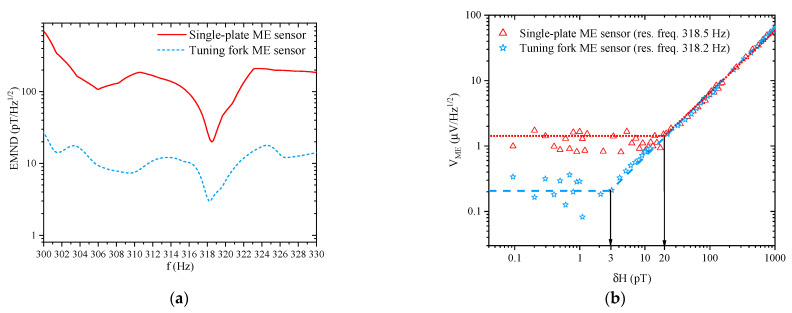
(**a**) Equivalent magnetic noise density dependences vs. frequency are presented in the range from 300 to 330 Hz for the single-plate and tuning fork ME sensors with the applied optimal bias H fields. (**b**) Linear output ME voltage response vs. applied AC magnetic field δH varying from 0.1 to 1000 pT at the resonance frequencies with the applied optimal bias H fields for the single-plate and tuning fork ME sensors. Reprinted with permission from [62]. Copyright 2019 Elsevier.

**Figure 15 sensors-21-06232-f015:**
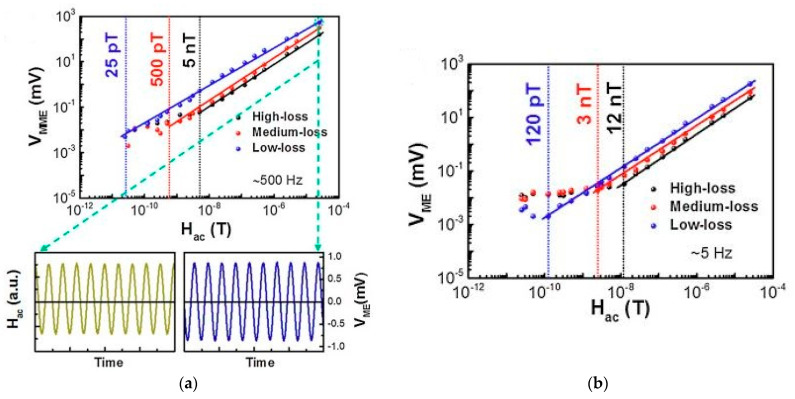
Magnetic sensitivity of the ME composite magnetic sensors under zero magnetic bias at (**a**) 500 Hz, and (**b**) 5 Hz. In the bottom panel of (**a**), the voltage output response of the low-loss composite sensor under the magnetic detection field of 25 × 10^−6^ T. Reprinted with permission from [28]. Copyright 2017 Elsevier.

**Figure 16 sensors-21-06232-f016:**
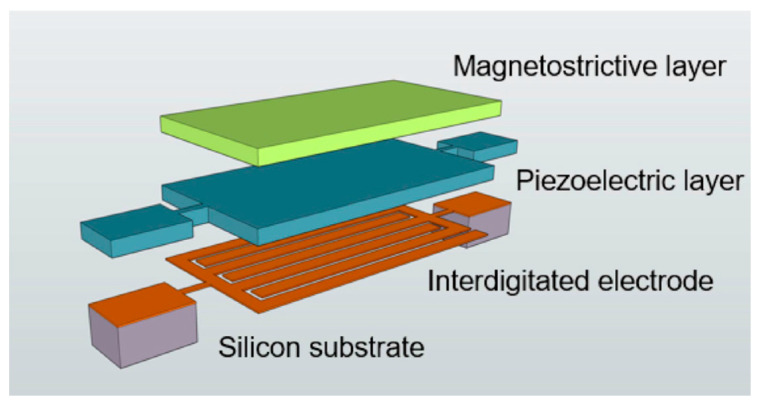
3D schematic of NEMS AlN resonant magnetic sensor. Reprinted with permission from [24]. Copyright 2017 AIP Publishing.

**Figure 17 sensors-21-06232-f017:**
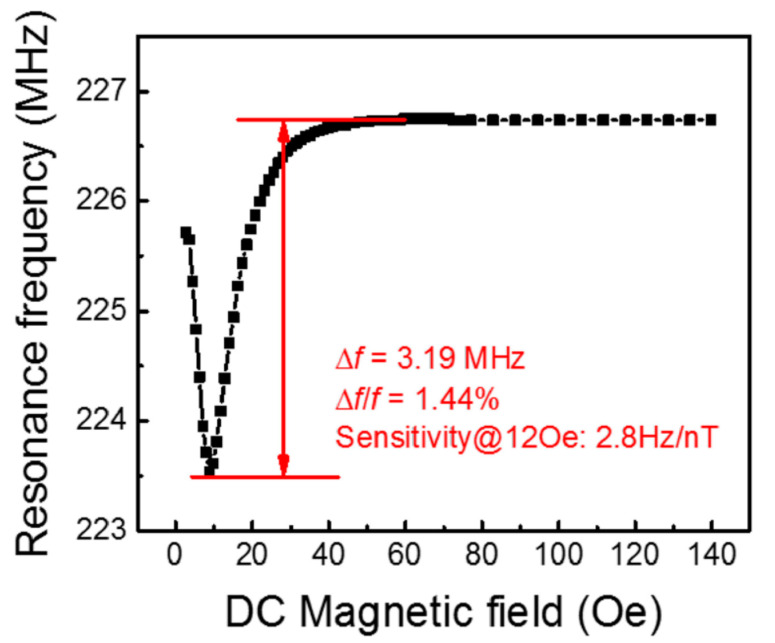
The frequency dependence of the ME sensor on the DC magnetic field. Reprinted with permission from [24]. Copyright 2017 AIP Publishing.

**Table 1 sensors-21-06232-t001:** The materials parameters of ME material components.

Material	s_11_ (10^−12^ m^2^/N)	s_12_ (10^−12^ m^2^/N)	s_13_ (10^−12^ m^2^/N)	s_33_ (10^−12^ m^2^/N)	q_11_ (10^−12^ m/A)	q_12_ (10^−12^ m/A)	d_31_ (10^−12^ m/V)	d_33_ (10^−12^ m/V)	ε_33_/ε_0_ μ/μ_0_	ρ (g/cm^3^)
PZT	15.3	−5	−7.22	17.3	–	–	−175	400	1750	7.75
PMN-PT	23	−8.3			–	–	−600		4100	8.1
AlN	2.9	−0.93	−0.5	2.8	–	–	−1.92	4.96	8.5	3.255
Terfenol-D	33.3	−10			15,707	4730	–	–	5	9.25
Ni	20	−7			−4140	1200	–	–	300	8.902
Metglas	10	−4			11,000	−2000	–	–	10^5^	7.6

**Table 2 sensors-21-06232-t002:** The material parameters of ME material components.

Material	Characteristic	Value
b-LN Y +140°	d/ε	0.5039 nC/N
b-LN Y +128°	d/ε	0.5534 nC/N
Metglas 2826 MB	Pseudopiezomagnetic coefficient q	4 ppm/Oe

**Table 3 sensors-21-06232-t003:** The parameters of different ME magnetic field sensors.

Structure Configuration	Equivalent Magnetic Noise Density	α_E_, V/(cm·Oe)	AC Magnetic Field Frequency	LoD	Reference
T-D/PZT/T-D C-C mode	20 pT/√Hz at 1 Hz	0.24	0.5 Hz–2 kHz	–	[31]
T-D/PZT	2 pT/√Hz at 1 Hz	0.12	133–145 kHz	–	[33]
Ni/PZT/T-D	2 pT/√Hz at 1 Hz	44.8	35.8 kHz	–	[35]
Metglas/interdigited electrodes/PMN-PT fibers	5.1 pT/√Hz at 1 Hz	52	1 kHz	–	[18]
FeCoBSi/AlN	5.4 pT/√Hz at 1 Hz	1200	330 Hz	–	[36]
Metglas/PMN-PT	10.8 pT/√Hz at 1 Hz	19.5	–	–	[37]
Metglas/PZT	40 pT/√Hz at 1 Hz	21.6	–	–	[38]
Metglas/PZT	15 pT/√Hz at 1 Hz	25	–	–	[39]
Metglas/PZT	9.8 pT/√Hz at 1 Hz	31.4	–	–	[40]
Metglas/PMN-PT	6.2 pT/√Hz at 1 Hz (≤1 pT/√Hz at 10 Hz)	61.5	–	–	[19]
Metglas/PZT	1–100 pT/√Hz at 1 Hz	–	–	–	[7]
Metglas/PZT	–	280.5	59 kHz	200 pT	[46]
Metglas/PMN-PT	20 pT/√Hz at 200 Hz	8.5	–	0.6 nT	[47]
Metglas/PMN-PZT	–	–	–	115 pT	[48]
Metglas/PZT	9.1 pT/√Hz at 1 Hz	–	–	2 pT	[49]
Metglas/PMN-PT	–	7000	23.23 kHz	135 fT	[50]
Head-to-head bidomain LN y +140°/Metglas 2826 MB (in resonance)	92 fT/Hz^1/2^ at 6862 Hz	1704	6862 Hz	200 fT	[60]
Bidomain LN y +128° tuning fork/Metglas 2826 MB (in resonance)	3 pT/Hz^1/2^ at 318.2 Hz	144.4	318.2 Hz	3 pT	[61]
Head-to-head bidomain LN y +128°/Metglas 2826 MB free mode (in resonance)	0.4 pT/Hz^1/2^ at 3166 Hz	478	1335 Hz	–	[54]
Head-to-head bidomain LN y +128°/Metglas 2826 MB cantilever mode (out of resonance)	126 pT/Hz^1/2^ at 10 Hz	7.2	10 Hz		[54]
Head-to-head bidomain LN y +128°/Metglas 2826 MB cantilever mode (in resonance)	23 pT/Hz^1/2^ at 527 Hz	440	243.6 Hz		[54]
Tail-to-tail bidomain LN y +127°/Metglas 2826 MB (out of resonance)	153 pT/Hz^1/2^ at 1 kHz	1.88	1 kHz		[55]
Tail-to-tail bidomain LN y +127°/Metglas 2826 MB (in resonance)	524 fT/Hz^1/2^ at 30.82 kHz	462.7	30.82 kHz		[55]
FeCoBSi/AlN	1 nT/√Hz at 1 Hz	583	669 Hz		[62]

## Data Availability

Not applicable.

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
