# Peer review of "Magnetoelectric Magnetic Field Sensors: A Review"

_sensors, 2021, doi:10.3390/s21186232_

Round 1

Reviewer 1 Report

This paper reports the parameters of magnetoelectric magnetic field sensors. This work contains many bibliographical references that make it interesting to know the progress in this field of science. However, this article is difficult to read, the technical data are poorly summarized, and it appears difficult to understand the difference between each type of sensor. Some quantitative data could replace qualitative considerations to provide more support to the paper. Several paragraphs could be summarized and analyzed in more detail to reveal the benefit of the authors.

In the introduction, the bibliography contains very few recent works. Adding more recent publications will be valuable.

p.1 l 18 the sentence “the possibility of their use for solving applied problems” is unclear. What are the applied problems

p.1 l23 Meaning of SQUIDSs acronym, or at least its function, has to be reminded

p.1 l.30-40 references have to be added to attest to this idea

p.2 l.54 the “significant results” and “high parameters” are not defined. This way, this sentence is not usable.

p.3 l. 97 “from picotestla and higher”. Higher is very qualitative and brings no real quantitative information.

p.3 fig 1 the quality of the figure is very low. This could easily be performed.

  1. 5 Table 1. This table is interesting to figure the choice of the material. Why are the properties of the materials used in the others sensors not detailed? A complete table of the properties of the material used in the mentioned sensors will help the reader understand the choice of materials. Some information on the maximum capacities (elongation for example) could be helpful. A discussion based on this complete table will provide interesting information to the reader.

Paragraph 2.2 l.180-450 this part is very well documented, but everything is complicated to read and to understand what is essential in the whole text. It looks more like a bibliographic study than a review paper. A summary or a different presentation based on figures or graphical representation would help clarify the message.

Table 3  The table format makes the data difficult to read and analyze. A more compact format will help the reader to use this interesting table.

Author Response

Thank you for your comments. Please see my replies in the PDF file.

Reviewer 2 Report

  1. In Section 2.3,it is mentioned that the quasistatic method is measured at 1kHzfrom lines 463-465, but in lines 469-470 it is said that the quasistatic method is used for measurement in a DC field, and the dynamic method is used when the resonance frequency f=30.4 kHz. Here, more explanation is needed.
  2. In discussion,What does the “mechanical figure” refer to in line 661?
  3. 3. Some pictures are not clear, such as Figure 12, Figure 13, etc.

Author Response

Thank you for your comments. Please see my replies in the PDF file

Reviewer 3 Report

The reviewed article presents the progress made in the research of ME magnetic field sensors. The authors described in detail the material and structure system for the ME sensors, inculding Terfenol – PZT/ PMN-PT, Metgas PZT/PMN –PT, Metglas – Lithium Niobate. The crucial factors for determining the sensor characteristics are given and also the areas of their applications. The review is overall well written, of good organization, and will be of interests for the community studying the sensor ability of the ME composites and structures. I recommend its publication as it is.

Author Response

The authors are deeply grateful to reviewer 3 for the high assessment of our work.

Round 2

Reviewer 1 Report

This new version of the article presents a higher value and seems easier to read. This work is interesting as a report on the evolution of magnetoelectric magnetic field sensors. However, the analysis from the author and the effort to summarize the data is relatively weak. More effort on these two points would have added real worth to the paper.
